# The effect of mindfulness-based Tai Chi Chuan on mobile phone addiction among male college students is associated with executive functions

Jizhao Li[1], Dongling Wang[2], Shuang Bai[3]*, Wanjiao Yang[3]

**1** China Wushu School, Beijing Sport University, Beijing, China, **2** Center Laboratory, Peking University School and Hospital of Stomatology & National Center of Stomatology & National Clinical Research Center for Oral Diseases & National Engineering Research Center of Oral Biomaterials and Digital Medical Devices, Beijing, China. **3** School of Kinesiology and Health, Capital University Of Physical Education And Sports, Beijing, China

* baishuang@cupes.edu.cn

## Abstract

### Objective

Mindfulness-based Tai Chi Chuan (MTCC) have been shown to contribute to improvements in cognitive and executive functions. Changes in inhibition, an aspect of executive function, have been closely linked to mobile phone addiction. However, the relationship between these elements remains unclear. This study aims to investigate the effects of an 8-week MTCC intervention on executive function, mindfulness levels, and mobile phone addiction in male college students. Additionally, the study explores the role of executive function in improving mobile phone addiction through MTCC interventions.

### Methods

Sixty-six male college students were selected as research subjects and randomly divided into a control group (33) and an experimental group (33). The control group maintained their normal physical activity levels without any additional intervention. In contrast, the experimental group underwent 8 weeks of MTCC training. Mindfulness levels were assessed using the Mindful Attention Awareness Scale (MAAS), while mobile phone addiction was evaluated by the Mobile Phone Addiction Index (MPAI). The Flanker task, 1-back task, and More-Odd Shifting task were employed to evaluate inhibition, updating, and shifting aspects of executive function, respectively.

### Results

(1) The 8-week MTCC intervention significantly improved mobile phone addiction among male college students, with the intervention group showing a lower post-intervention MPAI score (46.09 ± 18.11) compared to the control group

**Data availability statement:** All relevant data are within the manuscript and its Supporting Information files.

**Funding:** This work was supported by the R&D Program of Beijing Municipal Education Commission (KM202210029001) and the Science and Technology School-strengthen Project of Capital University of Physical Education and Sports (155223005). The funders had no role in study design, data collection and analysis, decision to publish, or preparation of the manuscript.

**Competing interests:** The authors have declared that no competing interests exist.

($56.55 \pm 16.02$), yielding a mean difference of $-10.46$ (95% CI: $-18.92$ to $-1.99$, $p = 0.016$). Mindfulness levels also improved significantly ($p = 0.046$), as did specific sub-functions of executive function: inhibition correct rate ($p < 0.001$), inhibition response ($p = 0.001$), and shifting correct rate ($p = 0.001$). No significant effects were observed for updating correct rate ($p = 0.527$) or updating response ($p = 0.303$). (2) Mobile phone addiction indices were significantly correlated with changes in inhibition response ($r = 0.756$, $p = 0.000 < 0.01$), updating response ($r = 0.035$, $p = 0.045 < 0.05$), and shifting response ($r = 0.397$, $p = 0.022 < 0.05$). (3) Mindfulness levels and inhibition levels were significantly correlated ($r = 0.394$, $p = 0.023 < 0.05$). (4) Changes in inhibition within executive functions partially mediated the improvement of mobile phone addiction, with the direct effect (0.716) and mediating effect (0.483) accounting for 59.72% and 40.28% of the total effect (1.199), respectively.

## Conclusion

MTCC exercises significantly increase cognitive functions, leading to increased inhibition and attentiveness, and may be helpful in the prevention of addictions, including cell phone addictions.

---

## Introduction

Smartphones plays an increasingly important role in modern society. According to the data released in the 50th Statistical Reports on Internet Development in China, by June 2022, mobile Internet users scale up to 1.051 billion. Mobile Internet rate climbed to 99.6% [1]. The rate of short video user accounts for 91.5% of the overall scale of netizen. Research shows that young people are more dependent on mobile phones in their daily lives, especially college students [2]. The study shows that young people are more dependent on mobile phones in their daily lives, especially college students. The problem of excessive use of mobile phones has led to the problem of mobile phone addiction [3]. Mobile phone addiction (MPA) is a kind of addictive behavior in which the excessive use of mobile phones causes adverse physical and psychological consequences to individuals under the premise of non-substance addiction [4,5]. It is characterized by intense craving for mobile phones, loss of control, psychological dependence, and impaired life functioning [6,7]. A study reported that the prevalence of MPA among Chinese undergraduates is 21.3% [8].

The current intervention research in MPA is still in the theoretical development stage, mainly focusing on cognitive therapy and exercise therapy, etc. [9,10]. The experiment of Li Li by using cognitive interventions showed that both mindfulness therapy and focal resolution short-term case therapy had significant effects on mobile phone addiction. The mindfulness cognitive-behavioral group therapy had the best effect on the impulsivity of addicts [11]. Mindfulness as a positive personality trait can improve college students' mobile phone addiction [12]. The

reason may be that the higher mindfulness level is, the higher the attention level is, which means the individual is more likely to focus on present activities, reduce individual redundancy and increase control over negative thinking, ultimately to improve overall cognitive ability [13]. It is showed as the phenomenon that the higher the mindfulness levels of college students are, the lower their sense of craving for mobile phones and their propensity to become addicted to mobile phones [14]. And college students who regularly participate in physical activity may have higher mindfulness levels and tend to develop fewer mobile phone addiction problems. In the study of Kim [15] et al., it showed that 8 weeks of aerobic exercise was found to significantly reduce the smartphone addiction index in college students. Apart from the aerobic exercise, basketball and traditional Baduanjin exercises were found to be effective in reducing mobile phone addiction among college students and improving mental health in terms of reducing anxiety, loneliness and stress [16]. Tai Chi Chuan, an exercise that promotes mindfulness both mentally and physically, has been shown to be effective in reducing Internet addiction [17] and drug addiction [18]. The core mechanism of Tai Chi Chuan is consistent with the principles of mindfulness therapy, which emphasizes the unity of mind and body and place the psychological promotion effect on core position. Therefore, the mindfulness-based Tai Chi Chuan (MTCC) is also often used as a comprehensive intervention to improve stress and relieve anxiety [19,20]. The effects of MTCC intervention on addictive behaviors have been reported, but research on its effects and mechanisms on improving mobile phone addiction is still scarce.

Executive function is a set of cognitive processes that are essential for the cognitive control of behavior and the effective management of goal-directed activities [22]. Executive function includes working memory, cognitive flexibility, and inhibitory control [23]. There is more and more evidence proving that both acute and long-term MTCC intervention can improve executive function and cognitive performance in a variety of populations [24–28]. Executive function is more closely related to mobile phone addiction, among which inhibition, the function of allowing individuals to focus on task-relevant information and suppress irrelevant information, is most directly related to individual's addiction and health [29], as demonstrated in Fig 1. According to the study of GAO et al., it was found that individuals with lower inhibitory control were more likely to compulsively and excessively use mobile devices [30]. Inhibition deficits may lead to an inability to resist the urge to use mobile phones, even in inappropriate or harmful situations. Based on the above studies, there is a relationship between the MTCC intervention and mobile phone addiction as well as the mindfulness level and executive function. So this study aimed to investigate the following three questions: (1) the effect of MTCC intervention on college students' mindfulness level and executive function and (2) the effect of MTCC intervention on college students' mobile phone addiction (3) whether executive function plays a mediating role in the effect of MTCC intervention on mobile phone addiction.

## Research methods

### The subjects and groups

Male college students were recruited from a university in Beijing between September 11th and October 15th, 2022. Stratified random sampling (by academic year and major) was applied to the university's official registry, which included all full-time undergraduate male students aged 18–21. Only male students were included to eliminate potential confounding effects of menstrual cycle-related hormonal fluctuations on emotional outcomes. As shown in Fig 2, of 288 initially screened individuals, 216 (75%) were excluded due to MPAI scores ≤40, 3 for joint diseases, and 2 for psychiatric disorders, leaving 72 eligible participants. These 72 participants were randomly assigned to the experimental (n = 36) and control (n = 36) groups using a computer-generated sequence by an independent statistician. The sample size determination was informed by prior smartphone addiction intervention studies [31,32] and feasibility constraints. During the 8-week MTCC intervention (September 20th–November 15th, 2022), 6 participants (8.3%) dropped out (3 per group: experimental—1 injury, 2 scheduling conflicts; control—1 consent withdrawal, 2 illnesses), resulting in 33 participants per group (66 total) for final analysis. A post-hoc power analysis was conducted using GPower 3.1, based on the observed effect

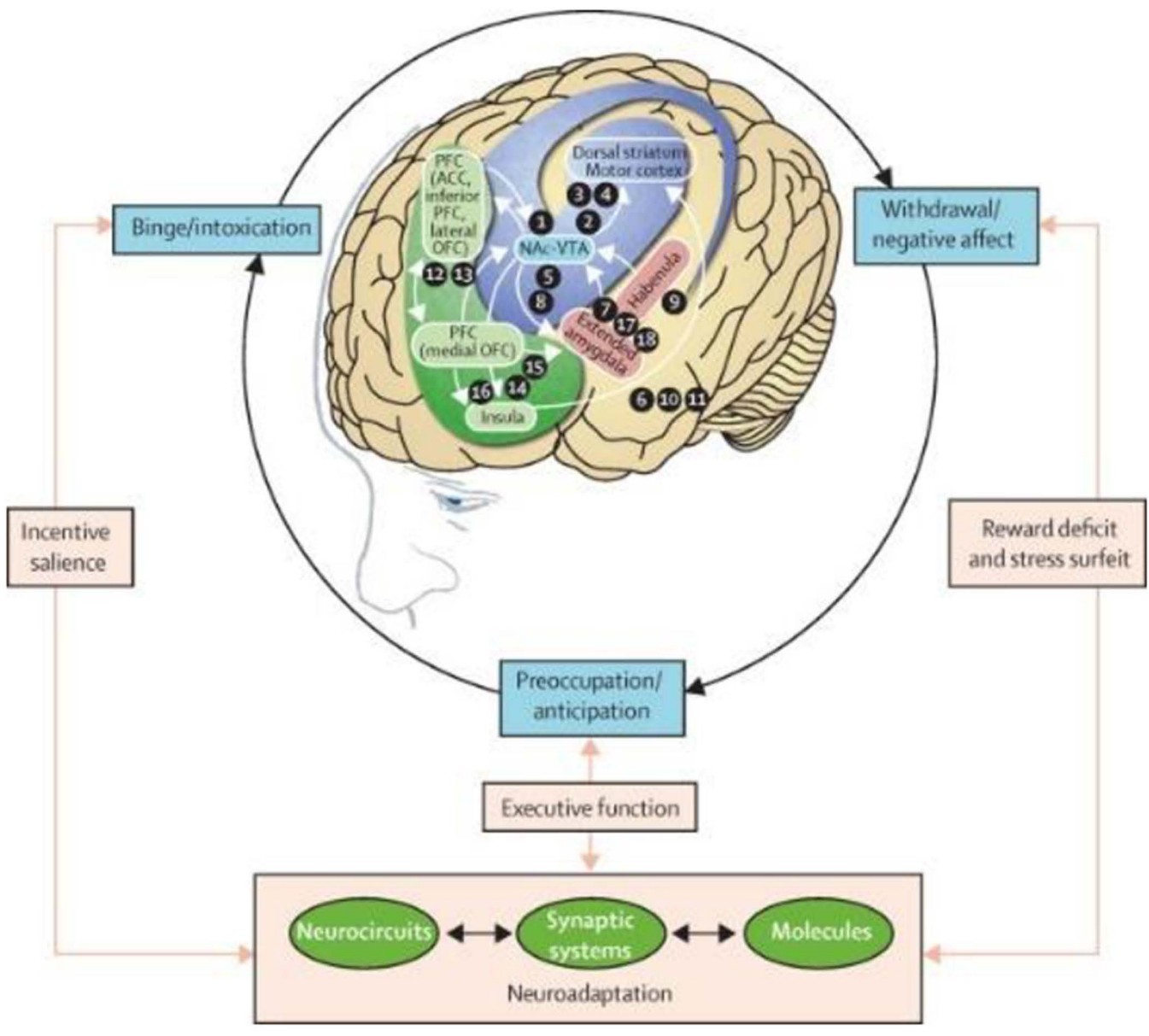

**Fig 1. Executive functioning is related to the reward effect of addictive behaviors [ 21].**

size (Cohen's d = 0.61) from the post-intervention data. With a sample size of 33 per group and α=0.05 (two-tailed), the achieved power was 84.5%, indicating that the study was adequately powered to detect the intervention effect.

Recruitment and informed consent were conducted by two research assistants blinded to group allocation. After baseline assessments, group assignments were revealed via sealed opaque envelopes managed by a third-party administrator. Due to the nature of the MTCC intervention, participants and intervention providers were not blinded. However, outcome assessors and data analysts remained blinded to group allocation throughout the study. MPAI scores were collected electronically using coded identifiers to maintain blinding.

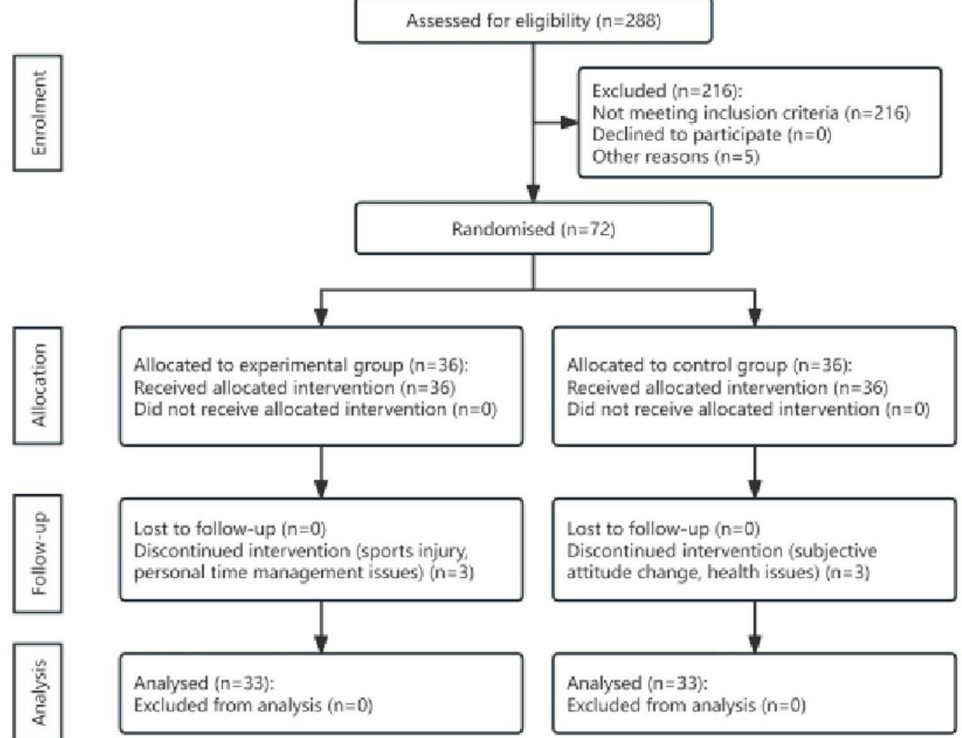

**Fig 2. CONSORT diagram of the trial participant flow.**

## Ethics statement

The human experiments conducted in this study were approved by the Biomedical Research Ethics Committee of the Capital University of Physical Education and Sports, No. 2021A27, and a paper ethical approval form was issued. All subjects have completed a paper version of the informed consent form for human experimentation. The study was conducted in strict adherence to the approved protocol, with no deviations throughout the trial. The trial was retrospectively registered on ClinicalTrials.gov (Identifier: NCT06837649; URL: https://clinicaltrials.gov/ct2/show/NCT06837649). The authors confirm that all ongoing and related trials for this intervention are registered.

## Exercise intervention program

Studies have shown that compared with the Tai Chi Chuan, MTCC is more relaxing to practitioner. The Tai Chi Chuan section is based on the Yang's movements [33], which not only conform to the basic technical characteristics, such as the steps, stance, techniques, movement of the center of gravity, and the coordination of the whole body between the movements, but also are clear, smooth, simple, easy to learn, and beautiful. The mindfulness section is complemented by soothing, soft music to guide the practitioner to focus on himself and return to inner peace. The essence of the program is to help practitioners focus their attention and achieve inner peace during practice. Through the practice of MTCC, practitioners are encouraged to face the difficulties they encounter in reality and to develop a positive and open mind to deal with negative emotions and thoughts in daily life. This program was designed without losing the ecological nature of the research results (see Table 1). The exercise time lasts 40 min each time, including 5 min of wuji pile practice with positive thoughts, 30 min of eight styles of Tai Chi Chuan practice (4 min*6 sets, 1 min interval between sets), and 5 min of relaxation part of wuji pile with music. The exercise frequency is 3 times a week.

**Table 1. Specific program for the 8-week MTCC intervention.**

| Intervention method | Mindfulness section | Tai Chi Chuan section[33] |
|---|---|---|
| Intervention content | Pre-intervention: Positive Mind Wuji Pile + soothing music + 3 min breathing space training<br>While intervention: Mindfulness Guidance + soothing Music + mindfulness Breathing<br>Post-intervention: Seated meditation + soothing music + 3 min breathing space training | Yang's eight styles of Tai Chi Chuan, including: (starting position, left and right rolled humerus, left and right knee stance, left and right wild horse mane, cloud hands, left and right standing on one leg, left and right foot stomp, left and right stroke the sparrow tail, cross hand, closing position)*6 |
| Intervention time | 10min | 30min (5min*6) |

## Measurement of positive thoughts, mobile phone addiction

The Mindful Attention Awareness Scale (MAAS) was used to measure the mindfulness level. It developed by Brown and Ryan in 2003 [34], which was introduced and revised by domestic scholars in 2012 [35]. Besides, it has high validity, structure validity and retest reliability of 0.87 0.89. The scale of single dimension scale, score from 1 to 6 points, are all positive ratings, which includes 15 problem. The higher the total score is, the higher levels one's mindful awareness and attention are.

Measurement of mobile phone addiction used the Mobile Phone Addiction Index (MPAI) [36] developed by Y.K. Leung, which is suitable for measuring mobile phone addiction in college students with a good reliability of 0.87. There are four dimensions in the scale, namely loss of control, withdrawal, avoidance, and ineffectiveness. It is rated from 1 to 5 with 17 questions.

## Measurement of executive function

Executive function was measured by the executive function test instrument [37], whose reliability has been tested in numerous studies. The test consisted of a Flanker task (inhibition function), a 2-back task (updating function), and a More-odd shifting task (shifting function). The items of test included response time (ms) and correct rate (%). Flanker task: the conditions were consistent like "FFFFF" while the conditions were not consistent like "FFLFF". Each condition was presented for 1000ms, with 500ms "+" presented in between. During the overall test, both conditions were presented randomly with equal probability. There were 12 times for practice before the formal data collection, and the formal task contained 48 judgments. The task required the subject to respond as quickly as possible while ensuring the correctness. The scores included the correct rate and response time (average response time for inconsistent conditions minus average response time for consistent conditions). The smaller the difference was, the better the inhibition function was.

## Mediating effects test method

A new mediating effect procedure proposed by Wen Zhonglin et al. was used to test for mediating effect [38]. The change of mindfulness level before and after the MTCC intervention and the change of mobile phone addition were extracted. A Pearson correlation analysis was conducted between the changes and the change of each sub-function of executive function respectively. With the Pearson correlation analysis, the relationship between the increase of mindfulness level, the decrease of mobile phone addiction and the change of executive function. The increase of the mindfulness level was used as the independent variable, and the decrease in the level of mobile phone addiction for which there was a correlation was used as the dependent variable. The mediating model 4 was selected using the Process plug-in. The mediating effect test was conducted using the value of the change in inhibition function in executive function, which had a correlation to determine the path of improvement of mobile phone addiction. Statistical analyses were all performed on SPSS 23.0 with a statistical analysis significance of $p < 0.05$. The mediating effect test was conducted using the bias-corrected percentile Bootstrap method, with 95% confidence intervals for the effect estimated by drawing a Bootstrap sample of 5000.

 

## Data statistics and analysis

The data were statistically analyzed by SPSS 23.0. Homogeneity tests were conducted using independent samples t-tests, and descriptive statistics were described in the form of average± standard deviation. To examine the effect of MTCC intervention mobile phone addiction, mindfulness level and executive function were as dependent variables, time (pre-test and post-test) as within-group factors and group (experimental and control groups) as between-group factors. A repeated measures analysis of variance was performed. By Pearson correlation analysis, the relationship between the change of mindfulness level (post-test vs pre-test), the change of mobile phone addiction (post-test vs pre-test) and the change of executive function (post-test vs pre-test). Mediation analysis was conducted using a path analysis framework with ordinary least squares (OLS) regression, following the causal steps approach [Baron & Kenny, 1986]. We employed the SPSS PROCESS macro (Model 4; Hayes, 2018] with 5,000 bootstrap samples to estimate direct and indirect effects. The model specified inhibition changes as the mediator between MTCC intervention (independent variable) and mobile phone addiction reduction (dependent variable). Bias-corrected 95% confidence intervals were calculated to assess significance.

## Results and analysis

### Homogeneity test of the indexes before intervention in the experimental and control groups

Independent samples t-tests were conducted on the pre-test data of mobile phone addiction index, mindfulness level, and each sub-function of executive function in the experimental and control groups. The results showed that mobile phone addiction index ($t = -0.110$, $p = 0.913$), positive thinking level ($t = 0.883$, $p = 0.381$), sub-functions of executive function of inhibition function (correct rate $t = -0.861$, $p = 0.393$; response time $t = -0.876$, $p = 0.384$), updating function (correct rate $t = -0.448$, $p = 0.656$; response time $t = -0.046$, $p = 0.963$) and shifting function (correct rate $t = 1.165$, $p = 0.248$; response time $t = 0.046$, $p = 0.963$). For all the pre-test data $p > 0.05$, indicating that the differences between the experimental and control groups were not statistically significant in terms of mobile phone addiction, mindfulness level and each sub-function level of executive function.

### Descriptive statistics of mobile phone addiction index of college students, mindfulness level and executive function before and after intervention

Descriptive statistics were conducted on the mobile phone addiction index, the mindfulness level and executive function of male college students before and after the MTCC intervention ( ±SD). Each sub-function of executive function was composed of two evaluations of response time and correct rate, where shorter response time stood for the better function, as shown in Table 2.

### Effects of MTCC intervention on mobile phone addiction index, mindfulness level and executive function among college students

A repeated measures analysis of variance was conducted using 2 (time: pre-test, post-test) × 2 (groups: experimental, control) to investigate the effects of MTCC intervention on each sub-function of mobile phone addiction index, mindfulness level and executive function among male college students, respectively.

### Effect of MTCC intervention on mobile phone addiction index and mindfulness level among male college students

The time main effect of mobile phone addiction index was highly significant [$F_{(1,65)} = 7.877$, $p = 0.007 < 0.01$], indicating that there was a tendency for mobile phone addiction index to change over time. The group main effect was not significant [$F_{(1,65)} = 2.553$, $p = 0.115 > 0.05$], indicating that there was no significant difference in mobile phone addiction index between

**Table 2. Descriptive statistics of the indicators of college students before and after MTCC intervention.**

| | | Experimental group ( ±SD) | | Control group ( ±SD) | |
|---|---|---|---|---|---|
| | | Pre-test | Post-test | Pre-test | Post-test |
| mobile phone addiction index | | 59.00 ± 19.14 | 46.09 ± 18.11 ## | 59.52 ± 18.83 | 56.55 ± 16.02 * |
| Mindfulness level | | 38.73 ± 11.16 | 43.76 ± 8.76 ## | 36.48 ± 9.40 | 37.12 ± 8.57 * |
| Executive Functions | Response time of inhibition | 10.83 ± 10.22 | 4.20 ± 10.42 ## | 12.85 ± 8.41 | 9.56 ± 10.38 ** |
| | Correct rate of inhibition | 0.93 ± 0.05 | 0.98 ± 0.02 ## | 0.95 ± 0.06 | 0.95 ± 0.04 ** |
| | Response time of updating | 1157.85 ± 269.75 | 1095.57 ± 208.01 | 1160.92 ± 270.94 | 1136.92 ± 309.77 |
| | Correct rate of updating | 0.76 ± 0.20 | 0.86 ± 0.12 # | 0.78 ± 0.15 | 0.80 ± 0.31 |
| | Response time of shifting | 309.75 ± 92.46 | 281.48 ± 71.12 | 308.66 ± 98.97 | 305.66 ± 93.95 |
| | Correct rate of shifting | 0.92 ± 0.17 | 0.95 ± 0.03 | 0.87 ± 0.20 | 0.87 ± 0.13 ** |

Note:

*indicates the significance of the posttest difference between the control group and the experimental group after the intervention, * indicates $p < 0.05$;

**indicates $p < 0.01$;

#indicates the significance of the posttest difference between the experimental group, # indicates $p < 0.05$;

##indicates $p < 0.01$. Values are presented as mean ± SD. For inferential statistics (mixed ANOVA, interaction effects, and difference-in-differences), refer to Figs 3-4.

the experimental and control groups. The time × group interaction effect was not significant [$F_{(1,65)}$ = 3.087, $p = 0.084 > 0.05$], indicating that there was no significant difference in the change of mobile phone addiction index between the two groups before and after the intervention.

Further simple effects analysis showed that there was a significant difference between the experimental group and the control group in the mobile phone addiction index in the post-test ($F = 6.167$, $p = 0.016 < 0.05$). There was a highly significant difference between the inhibition function in the pre-test and post-test of the experimental group ($F = 7.917$, $p = 0.006 < 0.01$). The mobile phone addiction index was very significantly lower after the intervention than before. The inhibition function in the pre-test and post-test of the control group. There was no significant difference between the pre-test and post-test inhibition functions of the control group ($F = 0.476$, $p = 0.493 > 0.05$).

The main effect of time on the mindfulness level was not significant [$F_{(1,65)}$ = 2.356, $p = 0.130 > 0.05$], indicating that there was no significant difference in the mindfulness level over time. The main effect of group was significant [$F_{(1,65)}$ = 5.708, $p = 0.02 < 0.05$], indicating that there was a significant difference in the mindfulness level between the experimental and control groups. The time × group interaction effect was not significant [$F_{(1,65)}$ = 3.919, $p = 0.052 > 0.05$], indicating that there was no significant difference in the change of the mindfulness level between the two groups before and after the intervention.

Further simple effects analysis showed that there was a significant difference in the mindfulness level between the experimental and control groups in the post-test ($F = 4.146$, $p = 0.046 < 0.05$). There was a highly significant difference in the mindfulness level between the experimental group in the pre-test and post-test ($F = 7.917$, $p = 0.006 < 0.01$), and the mindfulness level was significantly better after the intervention than before. There was no significant difference in the inhibition function between the control group in the pre-test and post-test. There was no significant difference between the pre-test and post-test inhibition function of the control group ($F = 0.083$, $p = 0.775 > 0.05$).

**Effects of the MTCC intervention on sub-functions of executive function among male college students**

The correct rate [$F_{(1,65)}$ = 9.457, $p = 0.003 < 0.01$] and response time of inhibition [$F_{(1,65)}$ = 32.073, $p = 0.000 < 0.01$] were highly significant on the time main effect, indicating a highly significant difference in inhibition function over time. The group main effect was not significant on the correct rate of inhibition [$F_{(1,65)}$ = 1.354, $p = 0.249 > 0.05$]. The group main effect

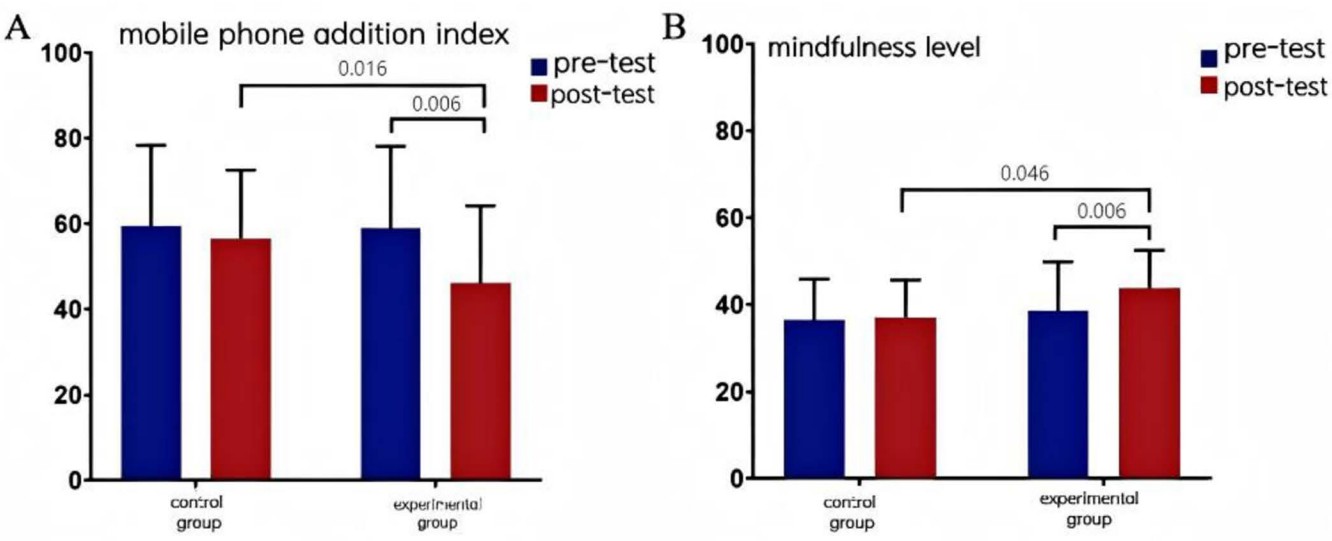

**Fig 3. Effects of MTCC intervention on mobile phone addiction index, mindfulness level and executive function of college students.**

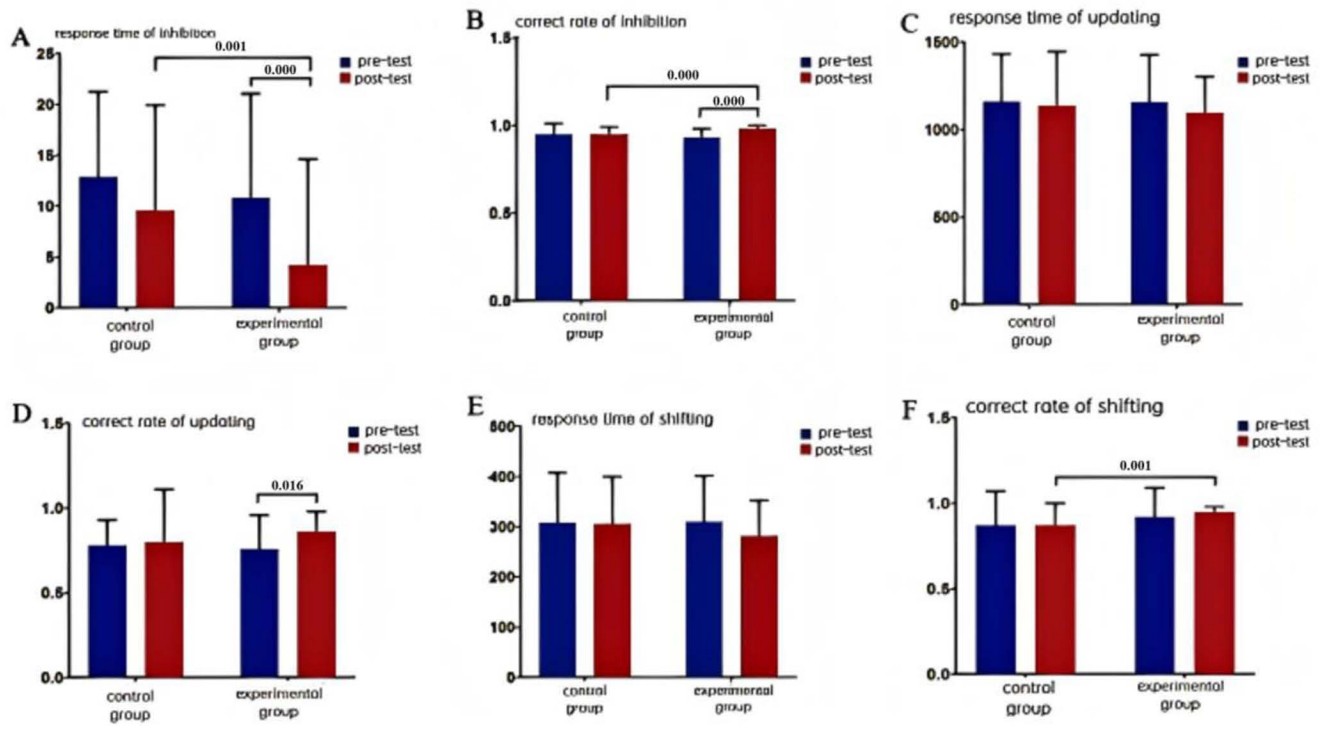

**Fig 4. Effects of MTCC intervention on sub-functions of executive function among male college students.**

was not significant in the correct rate of inhibition [$F_{(1,65)}$ = 7.305, $p$ = 0.009 < 0.01] and very significant in the response time of inhibition [$F$ = 7.305, $p$ = 0.009 < 0.01], indicating a significant difference in inhibitory between the experimental and control groups. The time × group interaction effect was very significant in the correct rate of inhibition [$F_{(1,65)}$ = 8.953, $p$ = 0.004 < 0.01] and response time of inhibition [$F_{(1,65)}$ = 8.885, $p$ = 0.004 < 0.01], indicating a significant difference in inhibition between the two groups before and after the intervention, indicating a highly significant difference in the change of inhibition between the two groups before and after the intervention.

Further simple effects analysis showed that post-test correct rate of inhibition ($F$ = 15.466, $p$ = 0.000 < 0.01) and response time of inhibition ($F$ = 13.330, $p$ = 0.001 < 0.01) were very significantly different between the experimental group and the control group. The correct rate ($F$ = 19.933, $p$ = 0.000) and the response time of inhibition ($F$ = 17.482, $p$ = 0.000 < 0.01) were very significantly different between the experimental group and the control group. The response time ($F$ = 17.482, $p$ = 0.000 < 0.01) was very significantly different, showing that the level of inhibition was very significantly better after the intervention than before. The correct rate ($F$ = 0.002, $p$ = 0.961 > 0.05) and response time of inhibition ($F$ = 2.010, $p$ = 0.161 > 0.05) measured before and after the control group were not significant difference.

The correct rate [$F_{(1,65)}$ = 3.018, $p$ = 0.087 > 0.05] and response time of updating [$F_{(1,65)}$ = 1.060, $p$ = 0.307 > 0.05] were not significant on the time main effect, indicating that the updating function did not differ significantly with time. The group main effect was not significant on the correct rate [$F_{(1,65)}$ = 0.296, $p$ = 0.588 > 0.05] and the response time of updating [$F_{(1,65)}$ = 0.192, $p$ = 0.663 > 0.05] were not significant, indicating that there was no significant difference between the experimental and control groups in updating function. The time × group interaction effect was not significant in correct rate [$F_{(1,65)}$ = 1.328, $p$ = 0.254 > 0.05] and response time of updating [$F_{(1,65)}$ = 0.209, $p$ = 0.649 > 0.05], indicating that there was no significant difference in the change of updating function between the two groups before and after the MTCC intervention.

Further simple effects analysis showed that there was no significant difference between the experimental group and the control group in the post-test correct rate ($F$ = 0.405, $p$ = 0.527 > 0.05) and response time of updating ($F$ = 1.078, $p$ = 0.303 > 0.05). There was a significant difference between the experimental group in the pre-test and post-test correct rate of updating ($F$ = 6.170, $p$ = 0.016 < 0.05). There was a significant difference in the pre-test and post-test correct rate of updating ($F$ = 6.170, $p$ = 0.016 < 0.05) in the experimental group, and the level of updating function was very significantly better after the intervention than before. While there was no significant difference at the response time of updating ($F$ = 1.103, $p$ = 0.298 > 0.05). There was no significant difference in the pre-test and post-test correct rate ($F$ = 0.113, $p$ = 0.738 > 0.05) and the response time of updating ($F$ = 0.112, $p$ = 0.739 > 0.05) in the control group.

The correct rate [$F_{(1,65)}$ = 0.345, $p$ = 0.559 > 0.01] and response time of shifting [$F_{(1,65)}$ = 0.981, $p$ = 0.326 > 0.05] were not significant on the time main effect, indicating that there was no significant difference in shifting function over time. The group main effect was not significant on the correct rate [$F_{(1,65)}$ = 6.783, $p$ = 0.011 > 0.05] and the response time of shifting [$F_{(1,65)}$ = 0.557, $p$ = 0.458 > 0.05], indicating that there was no significant difference in shifting function between the experimental and control groups. The time × group interaction effect was not significant in the correct rate [$F_{(1,65)}$ = 0.318, $p$ = 0.575 > 0.05] and the response time of shifting [$F_{(1,65)}$ = 0.642, $p$ = 0.426 > 0.05], indicating that there was no significant difference in the change of shifting function between the two groups before and after the intervention.

Further simple effects analysis showed that there was a highly significant difference between the experimental and control groups on the post-test correct rate ($F$ = 12.474, $p$ = 0.001 < 0.01) and the response time of shifting ($F$ = 1.309, $p$ = 0.243 > 0.05). There was no significant difference between the experimental group on the pre-test and post-test correct rate ($F$ = 0.958, $p$ = 0.331 > 0.05) and response time of shifting ($F$ = 1.938, $p$ = 0.169 > 0.05) were not significantly different. In the control group, there was no significant difference in the correct rate ($F$ = 0.000, $p$ = 0.989 > 0.05) and the response time of shifting ($F$ = 0.016, $p$ = 0.900 > 0.05) at pre-test and post-test.

   

### Relationship between changes in executive function and changes in mobile phone addiction index and changes in mindfulness level before and after the intervention

By Person correlation analysis, the correlations between the change in mobile phone addiction index, the change of mindfulness level and the change in correct rate and response time of each sub-function of executive function in the experimental group were examined including the correlations between the change in the mobile phone addiction index and the change in the response time of inhibition (r = 0.756, p = 0.000 < 0.01), the change in the response time of updating (r = 0.035, p = 0.045 < 0.05) and change in the response time of shifting (r = 0.397, p = 0.022 < 0.05) were significantly correlated. While change in mindfulness level was only significantly correlated with change in the response time of inhibition (r = 0.394, p = 0.023 < 0.05). As shown in Table 3.

### Mediating effects of executive function in the increase of mindfulness level and decrease of mobile phone addiction index before and after MTCC intervention

Based on the results of the two-by-two correlation between the decrease of the mobile phone addiction index, the increase of mindfulness level and the change in inhibition, respectively, with the change in inhibition as the mediating variable, the increase in mindfulness level as the independent variable, and the decrease in mobile phone addiction index as the dependent variable. Model 4 (simple mediation model) in the SPSS by Hayes (2012) was used [39] for mediating model testing. The results revealed that increased inhibition played a mediating effect in the relationship between increased mindfulness level and decreased mobile phone addiction index, as shown in Fig 5.

The specific process was that the predictive effect of increased mindfulness level on the decrease of mobile phone addiction index was significant (B = 1.199, t = 4.1892, p = 0.0002 < 0.01). The direct predictive effect of increased mindfulness level on the decrease of mobile phone addiction index remained significant when mediating variables were put in (B = 0.716, t = 3.1968, p = 0.033 < 0.05). The predictive effect of elevated mindfulness levels on changes in inhibition function was significant (B = 0.283, t = 2.388, p = 0.023 < 0.05), as was the positive predictive effect of elevated inhibition function on decreased mobile phone addiction index (B = 1.7081, t = 5.575, p = 0.000 < 0.01). In addition, the upper and lower limits of Bootstrap 95% confidence intervals for the direct effect of elevated mindfulness level affecting the decrease in mobile phone addiction index (95% CI: 0.258 to 1.173). The mediating effect of inhibition function change (95% CI: 0.069 to 0.998) did not contain 0, indicating that elevated mindfulness level not only directly predicted the decrease in mobile phone addiction index, but also was able to predict the decrease in mobile phone addiction index through the mediating effect of inhibition function changes. This direct effect (0.716) and mediating effect (0.483) accounted for 59.72% and 40.28% of the total effect (1.199) respectively, as shown in Table 4.

**Table 3. Correlation analysis of the change of executive functions with the decrease of mobile phone addiction index and the increase of mindfulness level.**

|  | the response time of inhibition | | the response time of updating | | the response time of shifting | |
|---|---|---|---|---|---|---|
|  | r | P | r | P | r | P |
| Decrease of mobile phone addiction index | 0.756 | 0.000** | 0.035 | 0.045* | 0.397 | 0.022* |
| Increase of mindfulness level | 0.394 | 0.023* | 0.083 | 0.645 | 0.066 | 0.716 |

Note:

*indicates p < 0.05;

**indicates p < 0.01

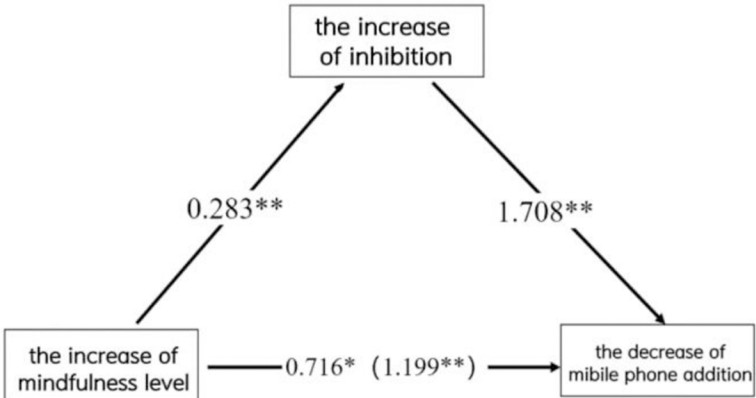

**Fig 5. Diagram of the mediating model of the relationship between increased inhibition in the increase of mindfulness level and the decrease of mobile phone addiction index.**

**Table 4. Results of the mediation model test for the elevated inhibition function in the relationship between elevated mindfulness level and decreased mobile phone addiction index.**

| Regression equation (N=33) | | Fitting index | | | Coefficient Significance | |
|---|---|---|---|---|---|---|
| Result Variables | Predictive variables | R | R2 | F (df) | B | t |
| The decline of mobile phone addiction index | Increase of mindfulness level | 0.601 | 0.362 | 17.55***(1) | 1.199 | 4.189** |
| | | | | | | |
| Changes of inhibition function | Increase of mindfulness level | 0.394 | 0.155 | 5.704*(1) | 0.283 | 2.388* |
| | | | | | | |
| | | | | | | |
| The decline of mobile phone addiction index | Increase of inhibition function | 0.825 | 0.681 | 31.963**(2) | 0.716 | 3.197** |
| | Increase of mindfulness level | | | | 1.708 | 5.475** |
| | | | | | | |

Note:

*$p < 0.05$,

**$p < 0.01$; all variables in the model were brought into the regression equation using unstandardized variables.

## Discussion

### Effects of the MTCC intervention on mobile phone addiction of college students and executive function

The results of variance in this study showed that an 8-week MTCC intervention significantly improved the mobile phone addiction index among male college students. Previous studies have discussed the effect aerobic exercise like running [40] or mind-body exercises like yoga [41] on mobile phone addition. They have shown that exercise is effective in improving mobile phone addiction and the severity of symptoms, as well as improving mental health. In addition, research has begun to explore the possibility of combining exercise interventions with other techniques, such as mindfulness practices. For example, Kim [42] conducted a study that combined yoga with mindful stress reduction to address mobile phone addiction among college students. The results of the study showed that the combined intervention significantly improved mobile phone addiction compared to the control group, suggesting that combining mind-body practices with mindfulness may improve the effectiveness of the intervention, which was consistent with this study.

Several potential mechanisms have been proposed in studies to explain the effects of physical and mental exercise on mobile phone addiction. First, both exercise and the use of mobile phone may activate the similar neurophysiological pathways in the brain. For example, studies have shown that mobile phone addiction affects brain regions associated with reward and loss of impulse control. Mobile phone addiction is associated with the release of dopamine in the reward system, which is in a similar manner to drug use [43]. Long-term exercise may help reduce addiction to mobile phones through its own effects on rewarding stimuli. Studies have shown that both forced and voluntary exercise increase reward-related neuroplasticity in key reward-based brain structures such as the dorsal striatum, ventral tegmental nucleus and ventral tegmental area [44]. Exercise has been shown to increase the release of endogenous opioids and endocannabinoids, which are associated with pleasure and reward. In addition, exercise has been found to improve cognitive function, particularly in the areas of attention and executive control, which may help individuals resist the compulsive urge to use mobile phones [45].

And the effect of exercise on executive function has been well documented in the literature. For example, Ludyga et al. [46] conducted a meta-analysis that found that motor interventions were associated with significant improvements in inhibition of executive function. Similarly, Álvarez-Bueno et al. [47] conducted a systematic review reporting that exercise also has a positive effect on inhibition, especially when the intervention involves the combination of cognitive training.

## Correlations between changes in executive function and changes in mindfulness level and mobile phone addiction before and after the MTCC intervention

The results of the correlational analysis showed a significant correlation between changes in executive function and changes in mindfulness level before and after the MTCC intervention, which is consistent with previous studies. Executive function is associated with many cognitive problems during the lifespan, including elevated rates of attention deficit hyperactivity disorder, depression, substance abuse, and antisocial behavior [48–50]. Increased mindfulness level allows individuals to focus more on awareness of thoughts, emotions, and behaviors, which can improve various aspects of executive function characteristics, including attention, cognitive control, and emotion regulation. The mechanism for this may be related to increased prefrontal cortex activity and changes in heart rate variability [51]. Specifically, neuroimaging studies have shown that inhibitory control and working memory, including attention control, attention shifting, cognitive flexibility and self-monitoring of potential responses, are associated with activation of the prefrontal cortex, specifically the anterior cingulate cortex, a key brain region in the prefrontal cortex associated with executive function. Tang et al. showed that increased mindfulness level led to corresponding changes in the prefrontal cortex (specifically the anterior cingulate cortex). This suggests a neurobiological mechanism behind the improvement of executive function [52]. Since changes in heart rate variability reflect autonomic activation, they also hypothesized that increased mindfulness level would show some effects on heart rate variability and thus on executive function.

And the relationship between changes in executive function and mobile phone addiction has also received attention from researchers, especially inhibition. Inhibitory control is one of the three core components of executive function[21]. Inhibitory control allows individuals to focus on task-relevant information and inhibit task-irrelevant information. In addition, inhibitory control helps individuals to control their emotions and behaviors [53,54]. Of the three core components of executive function (i.e., inhibitory control, working memory, and cognitive flexibility), inhibitory control is most directly related to a person's addiction or healthy behaviors. Research has shown that inhibitory control is an important predictor of healthy behaviors [52]. Neurobiological studies have shown a close relationship between the circuits underlying inhibitory control and the circuits disrupted by addictive behaviors. Indeed, research has suggested that deficits in inhibitory control are one of the key factors contributing to mobile phone addiction. Previous research has shown that individuals with higher levels of mobile phone addiction tend to have poorer inhibitory control. The increased inhibitory control is associated with lower levels of mobile phone addiction [30,54,55].

In conclusion, inhibition is an important cognitive strategy in mobile phone addiction [2]. The involvement of inhibition in the cognitive process is limited among mobile phone addicted people, while the increase of mindfulness level can improve the problem of inhibition and make individuals more inclined to use better self-control strategies.

### Mediating effects of executive function in MTCC intervention to improve mobile phone addiction

The results of the mediating effect test showed that the increase of inhibition after the 8-week MTCC intervention partially mediated the increase of mindfulness level and the decrease of mobile phone addiction, which is consistent with the results of previous studies.

Lan et al. [56] and Tang et al. [57] both observed the effects of increased mindfulness level on mobile phone addiction among college students and adolescents through a cognitive-behavioral mindfulness intervention, and found that changes in cognitive abilities, such as executive functions, mediated this process. The study by Tang et al. showed that children's brains are highly plastic and executive function is a key part during the process of brain development. Therefore, it has been suggested that executive abilities play an important role in sport to improve emotion regulation during the adolescent stage [58]. However, some studies have also shown that executive function plays a larger role in emotion regulation during childhood, adolescence, and early adult stages, allowing people to better use multiple modalities(e.g., distraction, mindfulness, reappraisal, etc.) to regulate [59]. By using the male as subjects, our pre-experiments found that exercise could similarly improve the rate of mobile phone addiction in adolescents through executive functions in the brain.

This suggests that this approach could not only indirectly improve the rate of mobile phone addiction by increasing inhibition, but also directly increase its effect on the rate of mobile phone addiction, although it is possible that there are other intermediate variables in this process. In addition, previous researchers have also suggested that changes in brain peptide hormones (NLRPs) occurred after the 8-week intervention, and that the endogenous production of peptide hormones in the body could improve patients' mindfulness level and emotional states, but their specific physiological functions are not yet clear, suggesting that NLRPs may also be an important mechanism that could improve patients' mindfulness level and mobile phone addiction.

This study for the first time combined Tai Chi Chuan and Mindfulness cognitive therapy. It explored the mediating effect of inhibition of executive function in improving mobile phone addiction after the comprehensive intervention, revealing the mediating mechanism by which intervention of the mindfulness level promotes the improvement of mobile phone addiction by affecting the level of inhibition. It providing a theoretical and practical basis for comprehensively revealing the intervention mechanism of mobile phone addiction. Future studies can continue to explore more mechanisms to provide theoretical and practical basis for the rational use of exercise to improve mobile phone addiction.

### Limitation

The absence of a priori power analysis led to sample size determination based on prior studies and feasibility considerations, though a post-hoc analysis confirmed adequate statistical power (84.5%) for the observed medium effect size (Cohen's $d = 0.61$). Additionally, the homogeneous sample—male students aged 18–21 from a single university in Beijing—limits generalizability to broader populations, including females, individuals from diverse age groups, and those in varied cultural or educational contexts. Future research should prioritize multi-center collaborations to enhance sample diversity and conduct a priori power calculations based on expected effect sizes to optimize methodological rigor and external validity.

### Conclusion

MTCC exercises significantly increase cognitive functions, leading to increased inhibition and attentiveness, and may be helpful in the prevention of addictions, including cell phone addictions.

## Supporting information

**S1 Data. Consort checklist of this research.**
(DOCX)

**S2 Data. Trial study protocol-Original Chinese version.**
(DOCX)

**S3 Data. Trial study protocol-English version.**
(DOCX)

**S4 Data. Original data.**
(XLSX)

## Acknowledgments

We sincerely thank all participants for their time and commitment to this study. Their contributions were essential to the successful completion of this research.

## Author contributions

**Formal analysis:** Shuang Bai.

**Funding acquisition:** Shuang Bai.

**Investigation:** Jizhao Li.

**Methodology:** Jizhao Li, Dongling Wang.

**Software:** Jizhao Li, Wanjiao Yang.

**Visualization:** Dongling Wang, Wanjiao Yang.

**Writing – original draft:** Jizhao Li, Shuang Bai.

**Writing – review & editing:** Dongling Wang, Wanjiao Yang.

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
