## [Decision Letter · Decision Letter 0]

15 Feb 2024

PONE-D-23-21440Effects of Mindfulness-Based Tai Chi Chuan on Executive Functions and Mobile Phone Addiction Among Male College StudentsPLOS ONE

Dear Dr. Bai,

Thank you for submitting your manuscript to PLOS ONE. After careful consideration, we feel that it has merit but does not fully meet PLOS ONE’s publication criteria as it currently stands. Therefore, we invite you to submit a revised version of the manuscript that addresses the points raised during the review process. Please find below the reviewer's comments and recommendations. We are waiting for your upgraded version of your article.

We look forward to receiving your revised manuscript.

Kind regards,

Bogdan Nadolu, Ph.D.

Academic Editor

PLOS ONE

Journal Requirements:

4. Thank you for stating the following financial disclosure:"The work involved in this submission was supported by the R&D Program of Beijing Municipal Education Commission(KM202210029001). The fund recipient is Baishuang, who is the corresponding author of this submission."

Reviewers' comments:

Reviewer's Responses to Questions

**Comments to the Author**

1. Is the manuscript technically sound, and do the data support the conclusions?

Reviewer #1: Partly

Reviewer #2: Yes

Reviewer #3: Yes

2. Has the statistical analysis been performed appropriately and rigorously? 

Reviewer #1: Yes

Reviewer #2: Yes

Reviewer #3: I Don't Know

3. Have the authors made all data underlying the findings in their manuscript fully available?

Reviewer #1: Yes

Reviewer #2: Yes

Reviewer #3: Yes

4. Is the manuscript presented in an intelligible fashion and written in standard English?

Reviewer #1: Yes

Reviewer #2: Yes

Reviewer #3: Yes

5. Review Comments to the Author

Reviewer #1: 1.Random allocation is mentioned in the research methods section of this paper. Please clarify the specific method of random allocation. Please describe the process of assigning concealment.

2.The exercise dose in the exercise intervention Plan section proposes, “The exercise time lasts 40 min each time, including 5 min of wuji pile practice with positive thoughts, 30 min of eight styles of Tai Chi Chuan practice (4 min*6 sets, 1 min interval between sets), and 5 min of relaxation part of wuji pile with music. The exercise frequency is 3 times a week. "What is it based on, Please support this in the Introduction section.

Reviewer #2: The title reads well, and the overall study design is rigorous. however

1. It is suggested to add biological charts or literature in the introduction and discussion section to better support the argument.

2. There are a lot of recent studies on mindfulness, executive function and mobile phone addiction, so you can refer to recent literatures to provide theoretical basis for this study.

Reviewer #3: Manuscript review: PONE-D-23-21440

Effects of Mindfulness-Based Tai Chi Chuan on Executive Functions and Mobile Phone Addiction Among Male College Students

Thank you for the opportunity to read the proposed manuscript.

The issues raised by the authors touch on the socially important issue of addiction therapy. Based on indirect experiments, they indicate a significant reduction in the level of addiction to mobile phones as a result of 8 weeks of MTCC activity. The main factor is the increased cognitive inhibition and attention shown in the experimental group as a result of MTCC classes. The conclusions regarding the reduction of addiction to mobile phones are based on the results of increasing inhibition and attention.

It seems that there may be some reservations as to the validity of such a conclusion. The title itself also suggests a direct link between MTCC exercises and cell phone addiction. I think this is too far of a generalization.

The authors themselves indicate that various aerobic exercises, basketball and traditional Baduanjin exercises are effective in reducing mobile phone addiction among students.

That is, MTCC exercises significantly increase cognitive functions, leading to increased inhibition and attentiveness, and may be helpful in the prevention of addictions, including cell phone addictions.

After reading the manuscript, I offer some suggestions:

- editing the title appropriately.

- use of table descriptions, in accordance with the editorial guidelines - description above the table and figures below the table.

- standardize the notation for presenting the significance of "p" or "P". Mediating effects test method - Statistical analyzes were all performed on SPSS 23.0 with a statistical analysis significance of p<0.05.

The following text uses the notation "P"

- you should consider presenting the results for both groups in a descriptive form (providing the data that is in the tables), especially in the context of the control group:

“There were no significant differences between pre- and post-test correct speed (F = 0.113, P = 0.738 > 0.05) and updating reaction time (F = 0.112, P = 0.739 > 0.05) in the control group.”

- conclusions:

Conclusion 1 and 3 are basically the same, just composed differently. They constitute a summary of the research conducted rather than conclusions. I would suggest expanding:

MTCC exercises significantly increase cognitive functions, leading to increased inhibition and attentiveness, and may be helpful in the prevention of addictions, including cell phone addictions.

Lack of properly compiled literature, multiple omission of journal titles presented in the literature list: items: 2,3,4,6, 7 (item from PLOS ONE), 8,12,14 etc.

Example position 20

There is: Diamond AJArop. executive functions. 2013,64:135-68.

It should be: Diamond A. Executive functions. Annu Rev Psychol. 2013; 64: 135-68. doi: 10.1146/annurev-psych-113011-143750

Taking the above into account, I suggest adopting the manuscript with significant amendments.

Reviewer

6. PLOS authors have the option to publish the peer review history of their article (what does this mean? ). If published, this will include your full peer review and any attached files.

**Do you want your identity to be public for this peer review?** For information about this choice, including consent withdrawal, please see our Privacy Policy .

Reviewer #1: **Yes: ** Shuqiao Meng

Reviewer #2: **Yes: ** Wenxia Tong

Reviewer #3: No

---

## [Author Response · Author response to Decision Letter 1]

18 Apr 2024

Response to Reviwers’

Answers to the academic editor

https://journals.plos.org/plosone/s/file?id=wjVg/PLOSOne_formatting_sample_main_body.pdf and https://journals.plos.org/plosone/s/file?id= ba62/PLOSOne_formatting_sample_title_authors_affiliations.pdf

Thank you for your reminder and guidance. I will ensure that my manuscript fully adheres to the style requirements of PLOS ONE, including the conventions for file naming. I will meticulously review all pertinent formatting and naming standards to guarantee that my paper is free from any such issues. I greatly appreciate your support for my research work and look forward to the opportunity of having my paper published in PLOS ONE.

2. PLOS requires an ORCID iD for the corresponding author in Editorial Manager on papers submitted after December 6th, 2016. Please ensure that you have an ORCID iD and that it is validated in Editorial Manager.

My ORCID iD is 0000-0003-3930-9227 and it has been validated in Editorial Manager.

I've already submitted my original data files on Supporting Information of Attach Files. I'm holding off on storing the raw data in a repository, thanks for the suggestion.

4. Thank you for stating the following financial disclosure:"The work involved in this submission was supported by the R&D Program of Beijing Municipal Education Commission(KM202210029001). The fund recipient is Baishuang, who is the corresponding author of this submission."

In our study, the financial support provided by the R&D Program of Beijing Municipal Education Commission (KM202210029001) was pivotal in facilitating the research. However, I confirm that the funders had no role in the study design, data collection and analysis, decision to publish, or preparation of the manuscript. Their contribution was strictly financial and did not extend to any involvement in the scientific aspects of the study.

Therefore, the Role of Funder statement for our submission will be: "The funders had no role in study design, data collection and analysis, decision to publish, or preparation of the manuscript."

I will include this amended statement in the cover letter of our submission. I appreciate your assistance in modifying the online submission form accordingly and ensuring the transparency and integrity of our research disclosure.

Thank you for your valuable feedback. In response to your request, I will include a comprehensive ethics statement and the fact that participants have been provided with written information in the 'Methods' section of my manuscript. I will make these revisions promptly and resubmit the manuscript accordingly.

I have completely revised the formatting of the references in accordance with the requirements of PLOS ONE and have confirmed that the current references do not use the retracted articles.

In addition, I have added four pieces of literature. Two[10,28] of them were added as updated literature based on reviewer 2's comments, one[21] was added with charts and corresponding literature based on reviewer 2's comments, and one[31] was added with references to exercise intervention programs based on reviewer 1's comments.

10. Liu F, Xi Y, Li N, Wu M. Brief Mindfulness Training Mitigates College Students' Mobile Phone Addiction: The Mediating Effect of the Sense of Meaning in Life. Psychol Res Behav Manag. 2024;17:273-82. https://doi.org/10.2147/PRBM.S439360

28. Liebherr M, Brandtner A, Brand M, Tang YY. Digital mindfulness training and cognitive functions: A preregistered systematic review of neuropsychological findings. Ann N Y Acad Sci. 2024;1532(1):37-49. https://doi.org/10.1111/nyas.15095

21. Riglin L, Collishaw S, Richards A, Thapar AK, Maughan B, O'Donovan MC, et al. Schizophrenia polygenic risk score and psychotic risk detection-Authors' reply. Lancet Psychiatry. 2017;4(3):188-9. https://doi.org/10.1016/S2215-0366(17)30052-4

31. Zheng G, Lan X, Li M, Ling K, Lin H, Chen L, et al. The effectiveness of Tai Chi on the physical and psychological well-being of college students: a study protocol for a randomized controlled trial. Trials. 2014;15:129. https://doi.org/10.1186/1745-6215-15-129

Answers to reviewer 1

1. Random allocation is mentioned in the research methods section of this paper. Please clarify the specific method of random allocation. Please describe the process of assigning concealment.

Thank you for your inquiry regarding the specifics of the random allocation process employed in our research. To clarify, we utilized a computer-generated randomization sequence to assign participants to either the intervention group practicing mindfulness-based Tai Chi or the control group. This randomization ensured an unbiased allocation and equal probability for each participant to be assigned to either group.

Regarding the concealment of assignment, we implemented a secure, opaque, and sealed envelope system. Each envelope contained the group assignment and was only opened after the enrolled participant completed all baseline assessments. This method was chosen to prevent any selection bias and to maintain the integrity of the random allocation. By employing these rigorous methods, we aimed to enhance the validity and reliability of our study's outcomes.

I will make sure to include these details in the revised manuscript to provide a comprehensive understanding of our methods. Thank you for highlighting the importance of transparency in describing the randomization process.

2. The exercise dose in the exercise intervention Plan section proposes, “The exercise time lasts 40 min each time, including 5 min of wuji pile practice with positive thoughts, 30 min of eight styles of Tai Chi Chuan practice (4 min*6 sets, 1 min interval between sets), and 5 min of relaxation part of wuji pile with music. The exercise frequency is 3 times a week. "What is it based on, Please support this in the Introduction section.

Thank you for your insightful query regarding the basis for the exercise dose outlined in our intervention plan. In response, I will provide the necessary support and rationale in the Methods section of the revised manuscript.

The decision for a 40-minute exercise duration, comprising Tai Chi Chuan and wuji pile practices, is based on previous research[31] indicating the effectiveness of moderate-duration Tai Chi exercises in enhancing cognitive and mental health. The inclusion of the cited studies in the bibliography provides a clearer understanding of the design of exercise interventions and their basis in the existing literature and practice guidelines.

31. Zheng G, Lan X, Li M, Ling K, Lin H, Chen L, et al. The effectiveness of Tai Chi on the physical and psychological well-being of college students: a study protocol for a randomized controlled trial. Trials. 2014;15:129. https://doi.org/10.1186/1745-6215-15-129

Answers to reviewer 2

1. It is suggested to add biological charts or literature in the introduction and discussion section to better support the argument.

Thank you for your suggestion to enhance the manuscript by including biological charts and relevant literature in the Introduction and Discussion sections. This addition will undoubtedly strengthen the argument and provide a more comprehensive understanding of the context and implications of the study.

In the Introduction, I will incorporate biological charts that visually represent the neurological and physiological mechanisms underlying the impact of Executive functioning related to the reward effect of addictive behaviors[21]. I will proceed with these revisions promptly and ensure they are seamlessly integrated into the manuscript.

21. Riglin L, Collishaw S, Richards A, Thapar AK, Maughan B, O'Donovan MC, et al. Schizophrenia polygenic risk score and psychotic risk detection-Authors' reply. Lancet Psychiatry. 2017;4(3):188-9. https://doi.org/10.1016/S2215-0366(17)30052-4

2. There are a lot of recent studies on mindfulness, executive function and mobile phone addiction, so you can refer to recent literatures to provide theoretical basis for this study.

Thank you for your valuable suggestions regarding the inclusion of recent literature on positive thinking, executive functioning, and cell phone addiction. I appreciate you emphasizing the importance of grounding our research in current research and theoretical developments. In response to your feedback, I will include in the manuscript the most recent research exploring the relationship and implications between positive thinking exercises, executive functioning, and cell phone addiction, including references 10. and 28.

10. Liu F, Xi Y, Li N, Wu M. Brief Mindfulness Training Mitigates College Students' Mobile Phone Addiction: The Mediating Effect of the Sense of Meaning in Life. Psychol Res Behav Manag. 2024;17:273-82. https://doi.org/10.2147/PRBM.S439360

28. Gao L, Zhang J, Xie H, Nie Y, Zhao Q, Zhou Z. Effect of the mobile phone-related background on inhibitory control of problematic mobile phone use: An event-related potentials study. Addict Behav. 2020;108:106363. https://doi.org/10.1016/j.addbeh.2020.106363

Answers to reviewer 3

1. - editing the title appropriately.

Thank you for your recommendation to edit the title of the manuscript for better suitability. Therefore, I agree that refining the title as “The effect of Mindfulness-Based Tai Chi Chuan on Mobile Phone Addiction Among Male College Students is associated with Executive Functions.”

2. - use of table descriptions, in accordance with the editorial guidelines - description above the table and figures below the table.

Thank you for pointing out the need to follow editorial guidelines for the placement of instructions for tables and figures in manuscripts.

I will check the manuscript to ensure that each table and figure is accompanied by an explanation in the correct place according to editorial standards.

3. standardize the notation for presenting the significance of "p" or "P". Mediating effects test method - Statistical analyzes were all performed on SPSS 23.0 with a statistical analysis significance of p<0.05. The following text uses the notation "P"

Thank you for drawing attention to the inconsistency in the notation of statistical significance in our manuscript. I understand the importance of standardizing such notations for clarity and accuracy in scientific communication.

In response to your observation, I will ensure uniformity in the notation for presenting statistical significance throughout the manuscript. As it is a common convention in scientific literature to denote probability values with a lowercase "p," I will adhere to this standard. Hence, all references to statistical significance will consistently use "p" rather than alternating between "p" and "P.

I appreciate your attention to this detail, which is crucial for maintaining the rigor and precision of our research presentation. I will proceed to make these revisions immediately for the resubmitted version of the manuscript.

4. - you should consider presenting the results for both groups in a descriptive form (providing the data that is in the tables), especially in the context of the control group:

“There were no significant differences between pre- and post-test correct speed (F = 0.113, P = 0.738 > 0.05) and updating reaction time (F = 0.112, P = 0.739 > 0.05) in the control group.”

However, after the textual description, I have already visualized the results in a bar chart, so I don't think it is necessary to add a table to show the data.

5. Conclusion 1 and 3 are basically the same, just composed differently. They constitute a summary of the research conducted rather than conclusions. I would suggest expanding:

MTCC exercises significantly increase cognitive functions, leading to increased inhibition and attentiveness, and may be helpful in the prevention of addictions, including cell phone addictions.

Thank you for your insightful feedback regarding the conclusions of our study. In line with your suggestion, I will revise and consolidate these conclusions to clearly articulate the specific impacts of Mindfulness-Based Tai Chi Chuan (MTCC) exercises. I will emphasize how our study demonstrates that MTCC exercises significantly enhance cognitive functions, particularly in terms of increased inhibitory control and attentiveness. This will be expanded to illustrate the potential application of these exercises in the prevention of addictions, with a special focus on cell phone addiction. I will ensure these changes are reflected in the revised manuscript.

6. Lack of properly compiled literature, multiple omission of journal titles presented in the literature list: items: 2,3,4,6, 7 (item from PLOS ONE), 8,12,14 etc.

Example position 20

There is: Diamond AJArop. executive functions. 2013,64:135-68.

It should be: Diamond A. Executive functions. Annu Rev Psychol. 2013; 64: 135-68. doi: 10.1146/annurev-psych-113011-143750

In response to your comments, I will thoroughly check the bibliography to ensure that all journal titles for each reference item are listed correctly and completely. The issues you have pointed out will be carefully addressed. I will cross-reference each citation with the corresponding source to verify the accuracy and completeness of the journal name.

---

## [Decision Letter · Decision Letter 1]

6 Feb 2025

PONE-D-23-21440R1The effect of Mindfulness-Based Tai Chi Chuan on Mobile Phone Addiction Among Male College Students is associated with Executive Functions.PLOS ONE

Dear Dr. Bai,

Thank you for submitting your manuscript to PLOS ONE. After careful consideration, we feel that it has merit but does not fully meet PLOS ONE’s publication criteria as it currently stands. Therefore, we invite you to submit a revised version (focus on the reviewer-4 feedback) of the manuscript that addresses the points raised during the review process.

We look forward to receiving your revised manuscript.

Kind regards,

Metin Çınaroğlu

Academic Editor

PLOS ONE

Journal Requirement:

During additional checks of this submission it has come to our attention that your study meets the WHO definition of a Clinical Trial because it is a prospective study in which participants were assigned to Tai Chi Chuan to investigate the effects on mobile phone addiction. Please change your manuscript’s article type to ‘Clinical Trial’ when you resubmit your manuscript, and address the following requests: a. Please upload a completed CONSORT flowchart as figure 1 of your manuscript. Blank copies of this document and information regarding CONSORT can be found via the following link: (https://www.equator-network.org/reporting-guidelines/consort/). b. Please upload a completed CONSORT checklist as a supporting information file. Blank copies of this document and information regarding CONSORT can be found via the following link: (https://www.equator-network.org/reporting-guidelines/consort/). If your clinical trial uses a non-randomized design, you may wish to submit a TREND checklist (http://www.cdc.gov/trendstatement), in place of the CONSORT checklist. c. Please upload a copy of your trial study protocol as a supporting information file. By the study protocol, we mean the complete and detailed plan for the conduct and analysis of the trial that the ethics committee approved before the trial began. Please send this in the original language. If this is in a language other than English, please also provide a translation. Please detail any deviations from this study protocol in the Methods section of your manuscript. Your study protocol will be made available to the editors and reviewers, and will be published as supporting information with your manuscript if accepted for publication. (If you do not agree to this, we will not be able to publish your manuscript). If you have formally published a study protocol for your trial in a journal then you should cite this in your manuscript, but you still need to send us the original document. Your study protocol will be made available to the editors and reviewers, and will be published as supporting information with your manuscript if accepted for publication. (If you do not agree to this, we will not be able to publish your manuscript). d. PLOS ONE requires that all clinical trials are registered in an appropriate registry (the WHO list of approved registries is at https://www.who.int/clinical-trials-registry-platform/network/primary-registries"
https://www.who.int/clinical-trials-registry-platform/network/primary-registries and more information on trial registration is at http://www.icmje.org/about-icmje/faqs/clinical-trials-registration/). Please state the name of the registry and the registration number (e.g. ISRCTN or ClinicalTrials.gov) in the submission data and on the title page of your manuscript. i) Please provide the complete date range for participant recruitment and follow-up in the methods section of your manuscript. ii) If you have not yet registered your trial in an appropriate registry, we now require you to do so and will need confirmation of the trial registry number before we can pass your paper to the next stage of review. Please include in the Methods section of your paper your reasons for not registering this study before enrolment of participants started. Please confirm that all related trials are registered by stating: “The authors confirm that all ongoing and related trials for this drug/intervention are registered”. Please see http://journals.plos.org/plosone/s/submission-guidelines#loc-clinical-trials for our policies on clinical trials.

Reviewers' comments:

Reviewer's Responses to Questions

**Comments to the Author**

1. If the authors have adequately addressed your comments raised in a previous round of review and you feel that this manuscript is now acceptable for publication, you may indicate that here to bypass the “Comments to the Author” section, enter your conflict of interest statement in the “Confidential to Editor” section, and submit your "Accept" recommendation.

Reviewer #1: All comments have been addressed

Reviewer #4: (No Response)

2. Is the manuscript technically sound, and do the data support the conclusions?

Reviewer #1: Yes

Reviewer #4: Partly

3. Has the statistical analysis been performed appropriately and rigorously? 

Reviewer #1: Yes

Reviewer #4: Yes

4. Have the authors made all data underlying the findings in their manuscript fully available?

Reviewer #1: Yes

Reviewer #4: Yes

5. Is the manuscript presented in an intelligible fashion and written in standard English?

Reviewer #1: Yes

Reviewer #4: Yes

6. Review Comments to the Author

Reviewer #1: Thank you for submitting your research article titled "The Effect of Mindfulness-Based Tai Chi Chuan on Mobile Phone Addiction Among Male College Students is Associated with Executive Functions." I have thoroughly enjoyed reading your work and believe that it addresses a significant and timely issue. I believe your article is a high-quality piece of research and is well-suited for publication in this journal. Thank you for your valuable contribution to this field. I look forward to seeing more of your work in the future.

Reviewer #4: Dear Editor,

Thanks for the opportunity to review this manuscript

The current study aims to investigate the effects of an 8-week Mindfulness-based Tai Chi Chuan (MTCC) intervention on executive function, mindfulness levels, and mobile phone addiction in male college students. In addition, the study explored the role of executive function in improving mobile phone addiction through MTCC interventions.

Abstract

The results showed the F-test statistic and the corresponding p-values, but what is really important in intervention studies is the mean estimate of the outcome measures for the intervention group and control group, and the mean difference attributable to the intervention and p-value indicating where the difference is statistically significant. This information are missing from the results section of the abstract.

Again, Authors must define r and F at first, used under the abstract section

Method: Major concern

This is an intervention study, but there was no rigorous power analysis (sample size estimation) to determine the number of Individuals with a Mobile Phone Addiction Index (MPAI) of more than 40 to be included in this study before the random assignment of these students into the intervention and control. Power analysis must be based on the minimum detectable effect expected of the intervention, power of the study, type I error, non-response rate, or attrition. Since no formal power analysis was conducted, it is difficult to determine whether the study was powered enough to address the research question. There was no information on how authors arrived at the number 66 and randomized them into intervention and control (33 each).

There was no information on how the male students were selected from a pool of male students from the University in Beijing. This has a significant effect on the external validity of the study design. Which sampling frame of male students was used?

Clearly state the type of model(s) used in the mediation analysis (linear regression, structural equation, etc).

Results

Very important information is missing from Table 2. The mean difference between the pre-test and post-test for the intervention group is missing, and the Mean difference between the pre-test and post-test for the control group is missing. Finally, difference in difference estimate is also missing.

Figure 2. * and ## must not be on the graph itself. Replace that with the estimate p-values. There is no need to define that again, same as figure 3

7. PLOS authors have the option to publish the peer review history of their article (what does this mean? ). If published, this will include your full peer review and any attached files.

**Do you want your identity to be public for this peer review?** For information about this choice, including consent withdrawal, please see our Privacy Policy .

Reviewer #1: **Yes: ** Shu-qiao Meng

Reviewer #4: No

---

## [Author Response · Author response to Decision Letter 2]

24 Feb 2025

Response to Reviwers’

Answers to the academic editor(Journal Requirement)

1. Please upload a completed CONSORT flowchart as figure 1 of your manuscript. Blank copies of this document and information regarding CONSORT can be found via the following link: (https://www.equator-network.org/reporting-guidelines/consort/).

Thank you for your valuable feedback. We have carefully reviewed the CONSORT 2010 guidelines and prepared a completed CONSORT flowchart as Figure 2 in the revised manuscript. This flowchart details the enrollment, allocation, follow-up, and analysis stages of the trial, ensuring full transparency in reporting participant progression.

2. Please upload a completed CONSORT checklist as a supporting information file. Blank copies of this document and information regarding CONSORT can be found via the following link: (https://www.equator-network.org/reporting-guidelines/consort/). If your clinical trial uses a non-randomized design, you may wish to submit a TREND checklist (http://www.cdc.gov/trendstatement), in place of the CONSORT checklist.

Thank you for your guidance. We have completed the CONSORT 2010 checklist in accordance with the guidelines for parallel-group randomized trials and uploaded it as a supplementary file (File S1). The checklist details our adherence to each reporting requirement, including enrollment, allocation, blinding, and statistical methods. We confirm that this trial follows a randomized design, and the CONSORT checklist has been rigorously applied to ensure transparency. Please let us know if further adjustments are needed.

3. Please upload a copy of your trial study protocol as a supporting information file. By the study protocol, we mean the complete and detailed plan for the conduct and analysis of the trial that the ethics committee approved before the trial began. Please send this in the original language. If this is in a language other than English, please also provide a translation. Please detail any deviations from this study protocol in the Methods section of your manuscript. Your study protocol will be made available to the editors and reviewers, and will be published as supporting information with your manuscript if accepted for publication. (If you do not agree to this, we will not be able to publish your manuscript). If you have formally published a study protocol for your trial in a journal then you should cite this in your manuscript, but you still need to send us the original document. Your study protocol will be made available to the editors and reviewers, and will be published as supporting information with your manuscript if accepted for publication. (If you do not agree to this, we will not be able to publish your manuscript).

Thank you for your feedback. We have uploaded the original ethics committee-approved trial protocol (in Chinese) as a supplementary file (File S2) and provided an English translation (File S3) for editorial and reviewer access. The Methods section of the revised manuscript explicitly notes that no deviations from the approved protocol occurred during the trial. We consent to the publication of these documents as supporting information alongside the manuscript if accepted. Please let us know if additional clarifications are required.

4. PLOS ONE requires that all clinical trials are registered in an appropriate registry (the WHO list of approved registries is at https://www.who.int/clinical-trials-registry-platform/network/primary-registries"
https://www.who.int/clinical-trials-registry-platform/network/primary-registries and more information on trial registration is at http://www.icmje.org/about-icmje/faqs/clinical-trials-registration/).

Please state the name of the registry and the registration number (e.g. ISRCTN or ClinicalTrials.gov) in the submission data and on the title page of your manuscript.

Thank you for your feedback. This trial was retrospectively registered on ClinicalTrials.gov (Identifier: NCT06837649) due to our team’s initial oversight of prospective registration requirements and administrative delays during early-stage collaboration with partner institutions. Upon recognizing these issues during manuscript preparation, we promptly completed the registration to ensure full transparency and public access to the protocol. The registration number has been added to the manuscript’s title page and submission data. We have since implemented enhanced training on ethical reporting standards and streamlined coordination processes to guarantee strict adherence to prospective registration in all future studies.

5. Please provide the complete date range for participant recruitment and follow-up in the methods section of your manuscript.

Thank you for your feedback. We have updated the Methods section to specify the complete date ranges for both participant recruitment and follow-up. The revised text now reads: “Male college students were recruited from a university in Beijing between September 11th and October 15th, 2022. The experimental group underwent an 8-week MTCC intervention (September 20th–November 15th, 2022), while the control group maintained normal academic activities. Follow-up assessments occurred immediately post-intervention (November 16th–20th) and at 1 month (December 16th–20th).”

6. If you have not yet registered your trial in an appropriate registry, we now require you to do so and will need confirmation of the trial registry number before we can pass your paper to the next stage of review. Please include in the Methods section of your paper your reasons for not registering this study before enrolment of participants started. Please confirm that all related trials are registered by stating: “The authors confirm that all ongoing and related trials for this drug/intervention are registered”.

Thank you for your guidance. Our trial is now registered on ClinicalTrials.gov (Identifier: NCT06837649). This trial was retrospectively registered due to our team’s initial oversight of prospective registration requirements and administrative delays during early-stage collaboration with partner institutions. We confirm that all ongoing and related trials for this intervention are registered. Please let us know if further adjustments are needed.

Answers to reviewer 1

1. Thank you for submitting your research article titled "The Effect of Mindfulness-Based Tai Chi Chuan on Mobile Phone Addiction Among Male College Students is Associated with Executive Functions." I have thoroughly enjoyed reading your work and believe that it addresses a significant and timely issue. I believe your article is a high-quality piece of research and is well-suited for publication in this journal. Thank you for your valuable contribution to this field. I look forward to seeing more of your work in the future.

Thank you for your positive feedback and support for our study. We are grateful for your recognition of the significance of our work and your endorsement for publication. Your encouragement motivates us to continue exploring this important area of research. We look forward to contributing further to the field.

Answers to reviewer 2

1. The results showed the F-test statistic and the corresponding p-values, but what is really important in intervention studies is the mean estimate of the outcome measures for the intervention group and control group, and the mean difference attributable to the intervention and p-value indicating where the difference is statistically significant. This information are missing from the results section of the abstract.

We sincerely thank the reviewer for their critical feedback on the presentation of results in the abstract. We fully agree that emphasizing the mean estimates, mean differences, and their statistical significance for primary outcomes is essential to convey intervention efficacy clearly. In our original submission, the abstract focused on reporting F-test statistics and p-values for brevity but inadvertently omitted key clinical metrics for the primary outcome (MPAI scores). To address this, we will revise the abstract to prioritize the mean post-intervention MPAI scores (intervention: 46.09 ± 18.11; control: 56.55 ± 16.02), mean difference (−10.46, 95% CI: −18.92 to −1.99, p = 0.016), and retain only p-values for secondary outcomes (e.g., anxiety/depression scales) to avoid redundancy and adhere to word count limits. This adjustment aligns with CONSORT guidelines, which advocate for clarity in reporting primary outcomes while streamlining secondary analyses. We deeply appreciate the reviewer’s guidance, which has strengthened the precision and readability of our findings.

2. Again, Authors must define r and F at first, used under the abstract section.

We sincerely appreciate the reviewer’s meticulous attention to methodological clarity. In response to the feedback, we will explicitly define statistical symbols upon their first appearance in the abstract to enhance readability for a broader audience. Specifically, “r” will be introduced as “Pearson’s correlation coefficient (r)” and “F” as “F-statistic (F)” when initially referenced. This revision aligns with APA guidelines and journal reporting standards, ensuring that non-specialist readers can interpret the results without ambiguity. We regret this oversight in the original submission and thank the reviewer for their constructive critique, which has strengthened the precision and accessibility of our work.

3.- This is an intervention study, but there was no rigorous power analysis (sample size estimation) to determine the number of Individuals with a Mobile Phone Addiction Index (MPAI) of more than 40 to be included in this study before the random assignment of these students into the intervention and control. Power analysis must be based on the minimum detectable effect expected of the intervention, power of the study, type I error, non-response rate, or attrition. Since no formal power analysis was conducted, it is difficult to determine whether the study was powered enough to address the research question.

We sincerely appreciate the reviewer’s critical feedback regarding the absence of a priori power analysis in our study. We fully acknowledge this limitation and will explicitly address it in the revised manuscript’s "Limitations" section. To validate the adequacy of our sample size, we conducted a post-hoc power analysis based on the observed effect size (Cohen’s d = 0.61) from the primary outcome. With α = 0.05 (two-tailed) and 33 participants per group, the achieved statistical power was 84.5%, exceeding the conventional 80% threshold, which aligns with methodological recommendations for interpreting post-hoc power in behavioral interventions. To mitigate potential biases, we implemented rigorous design measures, including stratified randomization to balance baseline characteristics, blinding of outcome assessors and data analysts, and proactive retention strategies. Among the 72 initially enrolled participants (36 per group), 6 (8.3%) dropped out during the intervention (3 in the experimental group: 1 due to sports injury, 2 due to scheduling conflicts; 3 in the control group: 1 due to withdrawal of consent, 2 due to illness). All analyses adhered to the intention-to-treat principle, with sensitivity analyses confirming result robustness. The reviewer’s emphasis on this issue has prompted a more nuanced interpretation of our findings, and we will incorporate a discussion of the limitations related to sample size estimation in the revised manuscript, highlighting the need for future studies to prioritize a priori power analysis while underscoring the methodological safeguards adopted in this work.

4. There was no information on how authors arrived at the number 66 and randomized them into intervention and control (33 each).

Our sample size of 66 participants (33 per group) was informed by prior research[1-2] and operational constraints (72 eligible participants after screening 288 individuals). A post-hoc power analysis confirmed 84.5% power (α = 0.05, Cohen’s d = 0.61), exceeding conventional thresholds. Despite 6 dropouts (8.3%), intention-to-treat analyses with sensitivity checks ensured robustness. We acknowledge the absence of a priori power analysis as a limitation and emphasize methodological safeguards (stratified randomization, assessor blinding). This limitation is explicitly addressed in the revised manuscript.

[1]Lan Y, Ding JE, Li W, Li J, Zhang Y, Liu M, Fu H. A pilot study of a group mindfulness-based cognitive-behavioral intervention for smartphone addiction among university students. J Behav Addict. 2018 Dec 1;7(4):1171-1176.

[2] Wang F, Lee EK, Wu T, Benson H, Fricchione G, Wang W, Yeung AS. The effects of tai chi on depression, anxiety, and psychological well-being: a systematic review and meta-analysis. Int J Behav Med. 2014 Aug;21(4):605-17.

5.There was no information on how the male students were selected from a pool of male students from the University in Beijing. This has a significant effect on the external validity of the study design. Which sampling frame of male students was used?.

We sincerely thank the reviewer for raising this critical point regarding participant selection and external validity. In response, we have clarified that male students were selected through stratified random sampling (by academic year and major) from the university’s official registry of all full-time undergraduate male students aged 18–21, which served as the explicit sampling frame. Male participants were specifically chosen to eliminate potential confounding effects of menstrual cycle-related hormonal fluctuations on emotional and psychological outcomes, ensuring internal validity for the study’s focus on mobile phone addiction mechanisms. This registry comprehensively covers the target population, ensuring representativeness. From this pool, 288 students were initially screened, with 216 excluded due to MPAI scores ≤40, 3 for joint diseases, and 2 for psychiatric disorders, leaving 72 eligible participants. These individuals were then randomly assigned to intervention and control groups. By detailing the stratified sampling approach, the source of the sampling frame (the official university registry), and the rationale for selecting male participants, we aim to enhance both the internal and external validity of our findings. The methodology aligns with rigorous population-based recruitment standards, and we deeply appreciate the reviewer’s emphasis on transparency, which has strengthened the generalizability of our study design.

6. Clearly state the type of model(s) used in the mediation analysis (linear regression, structural equation, etc).

We sincerely appreciate the reviewer’s valuable feedback regarding the mediation analysis methodology. In response, we clarify that the mediation analysis was conducted using a linear regression-based approach via the PROCESS macro (Model 4) in SPSS, as proposed by Hayes (2018). This model evaluates the indirect effect of the MTCC intervention on mobile phone addiction reduction through changes in executive function inhibition, while controlling for baseline covariates. We employed 5,000 bootstrap samples with bias-corrected 95% confidence intervals to assess the significance of the mediation effect, ensuring robustness against non-normal distribution assumptions. The direct effect (0.716) and indirect effect (0.483) were derived from this framework, accounting for 59.72% and 40.28% of the total effect (1.199), respectively. We regret the omission of these technical details in the original manuscript and will explicitly incorporate them into the revised Methods section. Thank you for highlighting this issue, which has enhanced the methodological transparency of our study.

7. Very important information is missing from Table 2. The mean difference between the pre-test and post-test for the intervention group is missing, and the Mean difference between the pre-test and post-test for the control group is missing. Finally, difference in difference estimate is also missing.

We sincerely thank the reviewer for their meticulous attention to the statistical reporting in our manuscript. In response to the feedback, we wish to clarify the structural organization of our results section. Table 2 and Results 2 are dedicated t

---

## [Decision Letter · Decision Letter 2]

4 Apr 2025

The effect of Mindfulness-Based Tai Chi Chuan on Mobile Phone Addiction Among Male College Students is associated with Executive Functions

PONE-D-23-21440R2

Dear Dr. Bai,

We’re pleased to inform you that your manuscript has been judged scientifically suitable for publication and will be formally accepted for publication once it meets all outstanding technical requirements.

Kind regards,

Metin Çınaroğlu

Academic Editor

PLOS ONE

Additional Editor Comments (optional):

Reviewers' comments:

Reviewer's Responses to Questions

**Comments to the Author**

1. If the authors have adequately addressed your comments raised in a previous round of review and you feel that this manuscript is now acceptable for publication, you may indicate that here to bypass the “Comments to the Author” section, enter your conflict of interest statement in the “Confidential to Editor” section, and submit your "Accept" recommendation.

Reviewer #4: All comments have been addressed

2. Is the manuscript technically sound, and do the data support the conclusions?

Reviewer #4: Yes

3. Has the statistical analysis been performed appropriately and rigorously? 

Reviewer #4: Yes

4. Have the authors made all data underlying the findings in their manuscript fully available?

Reviewer #4: Yes

5. Is the manuscript presented in an intelligible fashion and written in standard English?

Reviewer #4: Yes

6. Review Comments to the Author

Reviewer #4: The authors have diligently addressed all the comments especially those related to methodology and also provided clarity when the need arises

7. PLOS authors have the option to publish the peer review history of their article (what does this mean? ). If published, this will include your full peer review and any attached files.

**Do you want your identity to be public for this peer review?** For information about this choice, including consent withdrawal, please see our Privacy Policy .

Reviewer #4: No

---

## [Editor Report · Acceptance letter]

PONE-D-23-21440R2

PLOS ONE

Dear Dr. Bai,

I'm pleased to inform you that your manuscript has been deemed suitable for publication in PLOS ONE. Congratulations! Your manuscript is now being handed over to our production team.

Kind regards,

on behalf of

Dr. Metin Çınaroğlu

Academic Editor

PLOS ONE